# Trends in Occurrence and Phenotypic Resistance of Coagulase-Negative Staphylococci (CoNS) Found in Human Blood in the Northern Netherlands between 2013 and 2019

**DOI:** 10.3390/microorganisms10091801

**Published:** 2022-09-07

**Authors:** Matthijs S. Berends, Christian F. Luz, Alewijn Ott, Gunnar I. Andriesse, Karsten Becker, Corinna Glasner, Alex W. Friedrich

**Affiliations:** 1Certe Foundation, 9713 AV Groningen, The Netherlands; 2Department of Medical Microbiology and Infection Prevention, University Medical Center Groningen, University of Groningen, 9713 AV Groningen, The Netherlands; 3Institute of Medical Microbiology, University Hospital Münster, 48149 Münster, Germany; 4Friedrich Loeffler-Institute of Medical Microbiology, University Medicine Greifswald, 17489 Greifswald, Germany

**Keywords:** coagulase-negative staphylococci, AMR, antibiotic resistance, *Staphylococcus*, MDRE, *Staphylococcus haemolyticus*, *Staphylococcus epidermidis*, epidemiology, trends

## Abstract

Background: For years, coagulase-negative staphylococci (CoNS) were not considered a cause of bloodstream infections (BSIs) and were often regarded as contamination. However, the association of CoNS with nosocomial infections is increasingly recognized. The identification of more than 40 different CoNS species has been driven by the introduction of matrix-assisted laser desorption/ionization time-of-flight (MALDI-TOF) mass spectrometry. Yet, treatment guidelines consider CoNS as a whole group, despite increasing antibiotic resistance (ABR) in CoNS. This retrospective study provides an in-depth data analysis of CoNS isolates found in human blood culture isolates between 2013 and 2019 in the entire region of the Northern Netherlands. Methods: In total, 10,796 patients were included that were hospitalized in one of the 15 hospitals in the region, leading to 14,992 CoNS isolates for (ABR) data analysis. CoNS accounted for 27.6% of all available 71,632 blood culture isolates. EUCAST Expert rules were applied to correct for errors in antibiotic test results. Results: A total of 27 different CoNS species were found. Major differences were observed in occurrence and ABR profiles. The top five species covered 97.1% of all included isolates: *S. epidermidis*, *S. hominis*, *S. capitis*, *S. haemolyticus,* and *S. warneri*. Regarding ABR, methicillin resistance was most frequently detected in *S. haemolyticus* (72%), *S. cohnii* (65%), and *S. epidermidis* (62%). *S. epidermidis* and *S. haemolyticus* showed 50–80% resistance to teicoplanin and macrolides while resistance to these agents remained lower than 10% in most other CoNS species. Conclusion: These differences are often neglected in national guideline development, prompting a focus on ‘ABR-safe’ agents such as glycopeptides. In conclusion, this multi-year, full-region approach to extensively assess the trends in both the occurrence and phenotypic resistance of CoNS species could be used for evaluating treatment policies and understanding more about these important but still too often neglected pathogens.

## 1. Introduction

Sepsis is a syndrome of physiologic, pathologic, and biochemical abnormalities induced by bloodstream infections (BSIs). It is the most frequent cause of death in hospitalized patients and has been recognized by the WHO as a global health priority [1,2]. For years, coagulase-negative staphylococci (CoNS) were not considered a cause of BSIs and were often regarded as contamination [3]. Yet, it has been shown that CoNS can cause BSIs and a high mortality rate [4,5], especially in immunocompromised patients and newborns [6,7]. Moreover, CoNS have become increasingly associated with nosocomial infections [8]. This is attributed to (i) an increase of multimorbid and immunocompromised patients that are more prone to infections, (ii) the increased use of inserted foreign body material in modern medicine, and (iii) the property of CoNS to adapt molecularly to the hospital environment by diverging into new strains [8,9]. Specifically, *Staphylococcus epidermidis* and *Staphylococcus haemolyticus* are associated with sepsis caused by foreign-body-related infections (FBRIs), such as central line-associated BSIs and prosthetic joint infections [10].

At present, the CoNS group consists of more than 40 different species [11], currently affected by some reclassifications as well as reassignments to the newly established *Mammaliicoccus* genus [12]. In this study, all CoNS species have been included as described prior the taxonomic reassignments. The CoNS group is highly heterogeneous in its prevalence in humans and, more importantly, its antibiotic resistance (ABR) patterns. Zooming in CoNS at the species level is therefore useful to evaluate treatment options for CoNS causing BSI. Moreover, the clinical interpretation and relevance of BSIs caused by CoNS are dependent on the determination at the species level, since not all species in the CoNS group are pathogenic and associated with sepsis or (other) nosocomial infections [8,13]. While the microbiological diagnosis of BSIs has for decades been based on blood samples cultivated in automated blood-culture systems, molecular and mass spectrometry (MS) approaches enable more reliable microbiological diagnosis [14,15]. Since 2012, matrix-assisted laser desorption/ionisation time-of-flight (MALDI-TOF) MS has become a standard for the identification of bacterial species and has, together with sequencing approaches, led to a rapid discovery of new species compared to formerly used techniques [16,17]. Prior to the use of MALDI-TOF MS, identification of CoNS was primarily performed with biochemical and physiological tests, which yielded variable results, particularly in less prevalent species [17]. Examples include *S. warneri*, *S. auricularis*, *S. capitis*, and other CoNS species that primarily colonize the skin of animals or are found on food products [18]. In addition, several novel CoNS species have been described within the past decade [11]. Due to less specific traditional test techniques, previously reported prevalences and ABR patterns of specific species in the CoNS group may have been unreliable or under-evaluated. Consequently, identification using MALDI-TOF MS has become crucial to analyze species-specific ABR.

ABR is a global healthcare problem and of great concern in the antibiotic therapy of BSIs. This also applies to the CoNS group where multi-drug resistance is common in species circulating in hospitals [19]. The rise of beta-lactam resistance in CoNS species has led to vancomycin as a first-line therapy against CoNS-mediated BSI in many countries, even though information about the pharmacokinetics and pharmacodynamics (PK/PD) of vancomycin against CoNS is limited [5,20,21,22]. To assess the constant change of ABR in CoNS and population structures, geo-spatial and temporal analyses of ABR are required.

In the Netherlands, country-wide ABR analyses are used to develop antibiotic treatment guidelines by the Dutch Working Party on Antibiotic Policy (Stichting Werkgroep Antibiotica Beleid, SWAB) [22,23]. Their recommendations are based on NethMap, an annually released national report about ABR and antibiotic consumption by the Dutch National Institute for Public Health and the Environment (Rijksinstituut voor Volksgezondheid en Milieu, RIVM) [22]. However, this national report does not specify nor address ABR on a patient, hospital, or regional level.

Therefore, to inform clinical decision-makers, this cross-sectional retrospective study provides an in-depth ABR data analysis of all CoNS isolates found in human blood cultures from 2013 until 2019 in the Northern Netherlands that were determined by MALDI-TOF MS. We aim to evaluate the differences in the occurrence of CoNS species and their ABR patterns using a full-region approach.

## 2. Materials & Methods

### 2.1. Study Setting and Patient Cohort

This study was performed within the Northern Netherlands (Figure 1), a geographic region with 1.7 million inhabitants [24]. Its three provinces are similar in population density: Drenthe (492,167 inhabitants, 184/km^2^), Friesland (647,672 inhabitants, 183/km^2^) and Groningen (583,990 inhabitants, 243/km^2^) [24]. The study population consisted of 10,786 patients hospitalized with suspected BSI in 15 participating hospitals (14 secondary care, one tertiary care) located within this region between 1 January 2013 and 31 December 2019. All hospitals included at least one intensive care unit (ICU). There was no age restriction on including patients.

### 2.2. Microbiological and Demographic Data

All blood cultures were routinely drawn and analyzed at one of the three medical microbiological laboratories in the region (Izore, Friesland; Certe, Groningen and Drenthe; University Medical Center Groningen). After routine processing, isolates were included in the study if the species was characterized as a member of the CoNS group and antibiotic test results were available. In the study period, CoNS species were the most prevalent microorganisms isolated from blood and accounted for 27.6% of all available 71,632 blood culture isolates. The following variables were available for all isolates: date, name of laboratory, name of the hospital, age, gender, and ID of the patient and type of ward (ICU, clinical, outward). Genotypic data were not available for this study, as genotyping was not part of routine analysis.

### 2.3. Species Determination and Antibiotic Susceptibility Testing (AST)

Routine processing in the laboratories included the incubation of blood cultures, allowing the colourimetric detection of CO_2_ produced by growing microorganisms. Determination of the taxonomic species level was performed using MALDI-TOF MS. Two laboratories cultivated blood samples using the BacT/ALERT system (bioMérieux, Marcy-l’Étoile, France) and identified bacterial strains using the VITEK MS system (bioMérieux, France). One laboratory cultivated blood samples using the BACTEC (Becton Dickinson, Oxford, UK) and identified bacterial strains using the Microflex System (Bruker Corporation, Billerica, MA, USA). Since the databases of these proprietary systems are not publicly available, a qualitative assessment could not be attained, nor was this available in public literature.

AST was performed using the VITEK 2 Advanced Expert System after isolates were incubated on blood agar plates containing 5% sheep blood (BA + 5% SB). Two laboratories used the VITEK 2 P-586 cartridge and one laboratory used the VITEK 2 P-657 cartridge, which are both developed specifically for Gram-positive bacteria such as staphylococci. All results were authorized and validated by at least two laboratory technicians and one clinical microbiologist. Since different VITEK 2 cartridges were used, not all isolates were tested for all antibiotics analyzed in this study. Appendix A contains a full list of all included isolates and their respective AST results.

### 2.4. Selection of Bacterial Isolates

First isolates were determined and selected using the AMR package for R to exclude duplicate findings following the M39-A4 guideline by the Clinical Laboratory Standards Institute (CLSI) [25,26]. This guideline defines first isolates based on the species level per patient episode, regardless of body site and other phenotypical characteristics. The episode length for this study was defined as 365 days, resulting in the inclusion of a unique species once a year per patient.

In this study, several additions were made in extension to the CLSI guideline. As the CLSI guideline only considers the genus/species per episode, we investigated the added value to include changes in the ABR profile per genus/species and episode. For this purpose, we weighted the ABR profile of six preselected antibiotics, which were specifically chosen based on clinical relevance for Gram-positive bacteria, such as CoNS: erythromycin, oxacillin, rifampicin, teicoplanin, tetracycline, and vancomycin. Any change in these antibiotics from susceptible to resistant or vice-versa within the same species in the same patient within one episode was considered a ‘first weighted isolate’. ABR data analysis results per species were included if at least 30 first isolates were available following the current CLSI guideline [25].

### 2.5. EUCAST Rules and Antibiotic Resistance Data Analysis

European Committee on Antimicrobial Susceptibility Testing (EUCAST) rules were applied to the AST results including EUCAST Expert Rules (v3.1, 2016), EUCAST Clinical Breakpoint Interpretations (v10.0, 2020), and EUCAST rules for Intrinsic Resistance and Unusual Phenotypes [27,28]. All applied changes can be found in Appendix A. Resistance was defined as the number of isolates with an antibiotic interpretation of R (resistant) divided by the total number of susceptible (S or I) isolates, following the latest EUCAST guideline [28].

### 2.6. Statistical Analysis

All statistical analyses were performed using R v4.0.3, RStudio v1.4, and the AMR package v1.7.1 [26,29], which are all freely and publicly available. The AMR package provided a comprehensive solution to clean and analyze microbial and antimicrobial data. Specifically, antibiotic interpretation results were cleaned to only contain values “R”, “S”, or “I”, using AMR::as.rsi(). Antimicrobial resistance was determined with AMR::resistance(), which allowed to use the latest EUCAST guideline.

To test for linear trends, linear regression analyses were performed. Contingency tables were tested with Fisher’s exact test when the size was 2 × 2 and Chi-squared tests otherwise. For likelihood ratio tests, exact binomial tests were used. Outcomes of statistical tests were considered significant when *p* < 0.05.

### 2.7. Ethical Considerations

Ethical approval and informed consent were not required due to the retrospective observational nature of the study (METc 2021/343). All data were anonymized at the associated laboratories before analysis.

## 3. Results

### 3.1. Patients and Included Isolates

A total of 10,796 patients were included in this seven-year study (Table 1). The median age was 67 (IQR: 52–78) and 46.7% (n = 5040) of the patients was female. A total of 19,803 CoNS isolates were included, of which 14,992 isolates were used for ABR data analysis based on the “first weighted isolates” algorithm. A selection of first isolates using solely the CLSI guideline [25] would have yielded 12,971 isolates (−13.5%, *p* < 0.001). On ICUs, 25.7% of the first weighted isolates was found in males compared to 17.0% in females (*p* < 0.001). The number of ICU patients with CoNS compared to non-ICU patient with CoNS showed a significant difference between secondary care (17.5%, n = 1403) and tertiary care (24.4%, n = 670, *p* < 0.001). Yet, no significant difference was observed in the number of CoNS isolates found in ICU patients between secondary care (21.0%, n = 2191) and tertiary care (22.8%, n = 1034).

At total of 27 different species of the CoNS group were found within the isolate collection (Table 2), including seven isolates which have been reclassified to the genus *Mammaliicoccus* (i.e., *M. lentus* and *M. sciuri*) after the study period.

The top five species covered 97.1% (n = 14,560) of all first weighted isolates: *S. epidermidis* (n = 7260, 48.4%), *S. hominis* (n = 5033, 33.6%), *S. capitis* (n = 1395, 9.3%), *S. haemolyticus* (n = 612, 4.1%), and *S. warneri* (n = 260, 1.7%).

The remaining 432 isolates (2.9%) consisted of: *S. lugdunensis* (n = 91, 0.6%), *S. saprophyticus* (n = 45, 0.3%), *S. pettenkoferi* (n = 44, 0.3%), *S. cohnii* (n = 43, 0.3%), *S. caprae* (n = 40, 0.2%), and 17 other species (n = 169, 1.1%).

### 3.2. Occurrence of CoNS Species

The occurrence of CoNS species was stratified by type of care, type of hospital ward, geographic province, gender, and age (Figure 2). Age was grouped into five groups: 0–11, 12–24, 12–24, 25–54, 55–74, and 75 or more years. When stratifying by species level and the different types of care, the proportion of *S. epidermidis* among all CoNS isolates was 62.5% in tertiary care (n = 2834) versus 42.3% in secondary care (n = 4426; *p* = 0.049). Overall, *S. hominis* was less occurrent in tertiary care (20.3%, n = 919) than in secondary care (39.4%, n = 4114, *p* = 0.013), while the occurrence of other CoNS species was comparable between secondary and tertiary care. Yet, major differences in relative occurrence were observed between ICU and non-ICU status in secondary care. On secondary care ICUs, *S. epidermidis* accounted for 55.9% of all first weighted CoNS isolates found while on non-ICU wards this was 39.1% (*p* < 0.001). In contrast, *S. hominis* accounted for 25.7% on secondary care ICUs, while on non-ICU wards this was 43.3% (*p* < 0.001). Notably, *S. hominis* was found 105 times (7.53%) in children under the age of one.

Although all three provinces in the study region are similar in population density and gender distribution [24], major differences were observed in the occurrence of CoNS species between those provinces in secondary care. The occurrence of *S. epidermidis* among CoNS species in secondary care hospitals in Friesland was 38.7% in contrast to 43.7% and 45.9% in Drenthe and Groningen, respectively (*p* < 0.001). *S. hominis* was significantly more often found in secondary care hospitals in Friesland (45.9%) than in Drenthe (33.3%) and Groningen (36.0%) (*p* < 0.001). Drenthe and Groningen did not differ significantly in the occurrence of CoNS species in secondary care.

Overall, there was no significant change in species distribution over the years. Stratified by gender, a linear increase of *S. hominis* over time (*p* = 0.001) and a decrease of *S. epidermidis* (*p* = 0.005) was found in males. In females, the occurrence of *S. hominis* also increased over time (*p* = 0.008), but no decrease of *S. epidermidis* or any other species was observed. In age groups, no significant trends in occurrence were observed.

Interestingly, in the university hospital, *S. haemolyticus* was more occurrent on Hematology wards than on other wards (*p* < 0.0001, Table 3). Of all first *S. haemolyticus* isolates in this hospital, 52.7% (n = 154) was from the Hematology ward, while this was 21.8% for *S. epidermidis* (n = 617), 18.9% for *S. hominis* (n = 174), and 3.6% for *S. capitis* (n = 11). Other hospitals participating in this study did not contain a specific Hematology ward but rather had general Internal Medicine wards.

### 3.3. Definition of CoNS Persistence

In this retrospective study, it was impossible to differentiate between contaminated blood cultures and BSI-associated blood cultures, as clinical information was not available. Yet, to assess probable cases of BSIs caused by CoNS, we defined ‘CoNS persistence’ as a surrogate. CoNS persistence was defined by at least three positive blood cultures drawn on three different days within 60 days containing the same CoNS species within the same patient. In total, we identified 294 cases of CoNS persistence (Table 4). Aside from *S. massiliensis* that caused CoNS persistence in only one patient, the relatively most common causal agent of CoNS persistence was *S. haemolyticus* (5.8%, n = 32, *p* < 0.001), followed by *S. epidermidis* (3.7%, n = 212, *p* < 0.001) and *S. lugdunensis* (3.4%, n = 3, *p* = 0.46).

### 3.4. Antibiotic Resistance Data Analysis

Clinically relevant antibiotics and their respective ABR profiles were analyzed and compared for the top five CoNS species. Figure 3 shows time trends regarding the ABR profiles to ten different clinically relevant antibiotics, while Table 5 contains resistance percentages of all applicable combinations of species and antibiotic agents. In the following subsections, more details on occurrence and trends are provided per antibiotic class based on Figure 3 and Table 5. Comprehensive ABR analyses per species of all available variables can be found in Appendix A.

### 3.5. Beta-Lactams

Methicillin resistance was as high as 61.9% (n = 4135) in *S. epidermidis*, which was thus the proportion of MRSE (methicillin-resistant *Staphylococcus epidermidis*) among all *S. epidermidis* isolates in this study. Methicillin resistance in *S. haemolyticus* was even higher (72.1%, n = 403), but considerably lower in all other CoNS species (13.4–38.6%). Almost all *S. epidermidis*, *S. haemolyticus*, and *S. hominis* were resistant to benzylpenicillin. Resistance to amoxicillin/clavulanic acid was 72.9% (n = 3026) in *S. epidermidis*. *S. haemolyticus* showed a strong linear increase in amoxicillin/clavulanic acid resistance (*p* < 0.001) since 2013 with 87% resistance in 2019 (n = 61).

### 3.6. Glycopeptides

Vancomycin resistance was found in six *S. epidermidis* isolates (0.1%) and in one *S. hominis* isolate (0.0%). Half of all *S. epidermidis* isolates showed resistance to teicoplanin (50.5%, n = 2752), which increased over the seven study years (min-max: 44.8–54.5%, *p* = 0.001). An increase in teicoplanin resistance was observed in *S. haemolyticus* (min-max: 10.9–44.0%, *p* < 0.001). Teicoplanin resistance remained low in *S. capitis* (1.4%, n = 17), *S. hominis* (5.1%, n = 202), and *S. warneri* (9.6%, n = 22).

### 3.7. Macrolides

Erythromycin resistance was highest in *S. haemolyticus* (77.6%, n = 437), followed by in *S. epidermidis* (51.5%, n = 3471), *S. hominis* (45.7%, n = 2086), *S. warneri* (17.5%, n = 40), and *S. capitis* (11.0%, n = 136). Resistance to azithromycin and clarithromycin was equal to erythromycin resistance, due to EUCAST expert rules. However, resistance to clindamycin remained lower than resistance to erythromycin in all species: 45.6% (n = 253) in *S. haemolyticus* and 43.4% (n = 2910) in *S. epidermidis*, 29.6% (n = 1347) in *S. hominis*, 4.4% (n = 10) in *S. warneri*, and 10.8% (n = 132) in *S. capitis*.

### 3.8. Fluoroquinolones

The highest ciprofloxacin resistance was found in *S. haemolyticus* (66.4%; n = 374) and *S. epidermidis* (51.5%; n = 3468). Resistance to moxifloxacin was 26.4% (n = 24) in *S. haemolyticus* and less than 10% in all other species.

### 3.9. Other Antibiotics

Resistance remained low to rifampicin in *S. haemolyticus* (5.0%; n = 28) and *S. epidermidis* (4.5%; n = 300) and remained less than 0.6% in all other species. Linezolid resistance was 0.4% (n = 5) in *S. capitis*, 0.4% (n = 17) in *S. hominis*, 0.2% (n = 5) in *S. haemolyticus*, 0.1% (n = 5) in *S. epidermidis*, and absent in *S. warneri*. Mupirocin resistance was 14.8% in *S. epidermidis* (n = 987; of note: 166 additional isolates tested as “I”) and between 1.7% and 6.5% in other species.

### 3.10. Other Relevant Species

Resistance in *S. lugdunensis* (n = 82, sixth most occurrent species) remained generally low: 11.9% (n = 5) to amoxicillin/clavulanic acid, 7.3% (n = 6) to oxacillin, 4.8% (n = 4) to ciprofloxacin, 15.4% (n = 10) to tetracycline, 3.7% (n = 3) to teicoplanin, and no resistance was observed to rifampicin, linezolid, and vancomycin.

*S. saprophyticus* (n = 45, seventh-most occurrent species) showed no resistance to ciprofloxacin, teicoplanin, rifampicin, and vancomycin. Resistance to erythromycin was 15.4% (n = 6), to linezolid 7.9% (n = 3), and to oxacillin 16.2% (n = 6).

*S. pettenkoferi* (n = 44, eighth-most occurrent species) showed no resistance to gentamicin, tobramycin, linezolid, teicoplanin, or vancomycin but resistance to oxacillin was 40.4% (n = 14). Resistance to ciprofloxacin (8.1%, n = 3) and trimethoprim/sulfamethoxazole (2.7%, n = 1) remained low.

### 3.11. Effect of Patient Age Groups on Antibiotic Resistance in CoNS

Thirty microorganism-drug combinations were analyzed of which 13 showed a significant linear trend associated with age groups (Figure 4). In *S. epidermidis*, resistance to beta-lactam antibiotics was found to be lower in older patients (amoxicillin/clavulanic acid: *p* = 0.002; cefuroxime: *p* = 0.014). This was also observed in all aminoglycosides (e.g., gentamicin: *p* = 0.017; tobramycin: *p* = 0.009), except for kanamycin where higher age was associated with increasing resistance (*p* = 0.011). *S. epidermidis* was also less resistant to carbapenems in older patients (imipenem: *p* = 0.046; meropenem: *p* = 0.047). In *S. hominis*, similar trends were observed, although the effect of resistance to kanamycin was stronger (*p* = 0.006). *S. capitis* showed significantly more resistance to tetracycline (*p* = 0.022) in older patients.

## 4. Discussion

The present study provides a comprehensive data analysis of species in the CoNS group and their associated ABR patterns in a full-region approach using solely MALDI-TOF MS for discriminating CoNS species. We selected and analyzed a total of 14,992 first weighted CoNS isolates from 10,786 patients over seven years and identified significant differences in the trends of occurrence of the different CoNS species.

Before MALDI-TOF MS, CoNS were often reported without the species name as formerly used techniques were not able to reliably discriminate species [17]. The ratio of all CoNS species presented in the current study (Table 2) shows that five species accounted for 97.1% of all 27 found CoNS species with *S. epidermidis* accounting for the largest subgroup (48.4%, n = 7260). This distribution of species recovered from human specimens largely confirms results by previous reports [9,30].

While CoNS are one of the most common reasons for health care-associated, mostly catheter- or device-related bloodstream infections, they are also typical contaminants of blood cultures [31,32]. Apart from the most frequently detected CoNS species, i.e., *S. epidermidis* and *S. haemolyticus*, pathogenicity has not been well established widely due to the lack of data [8]. For this reason, we aimed at providing a pathogenicity assessment by defining CoNS persistence as a surrogate: at least three positive blood cultures drawn on three different days within 60 days, containing the same CoNS species. This definition was applied for two reasons. Firstly, it rules out contamination since the chance of finding the same contaminating species three times on three different days is expected to be low. Secondly, it prevents underestimating the possible pathogenicity of CoNS species since three sequential findings indicate CoNS persistence. Similar strategies have also been proposed by others [33]. In total, 294 different cases of CoNS persistence were identified (Table 4) among the 10,786 included patients. *S. haemolyticus* was found to be proportionally more associated with CoNS persistence (5.8%) than *S. epidermidis* (3.7%) and *S. hominis* (0.9%), although the latter two were 8–10 times more prevalent than *S. haemolyticus*. *S. epidermidis* has widely been recognized as a pathogen and an important cause of BSIs [5,34]. It was probably found more often than *S. haemolyticus* due to its stronger association with skin colonization [8], although we could not confirm this finding. It has been reported that *S. haemolyticus* is an emerging threat and one of the most frequent etiological factors of staphylococcal infections [9,35]. Most of the isolates of this species can produce biofilms, however, with differing biofilm structure compared to *S. epidermidis* [36]. CoNS species have also been found the most prevalent biofilm producers on hematological wards [37], which combined may explain the high occurrence of *S. haemolyticus* on these wards (Table 3). Recently, potent cytolytic activities toward human erythrocytes and leukocytes due to the possession of phenol-soluble modulins have been reported [38]. ABR in *S. haemolyticus* is of increasing concern and has been reported in a Brazilian study with 75% of analyzed *S. haemolyticus* isolates to be multi-resistant [35]. We confirmed this in the present study in which the ABR data analysis showed that 72.1% of *S. haemolyticus* isolates were resistant to oxacillin and 77.6% resistant to macrolides.

ABR data analysis also showed substantial differences between CoNS species (Figure 3, Figure 4, Table 5). This observation could be supported by another recent study from Brazil that showed strong heterogeneity in the resistance genes for CoNS species [39]. For example, the *blaZ* and *aac-aphD* genes that can lead to penicillin and aminoglycoside resistance, respectively, were found to be up to four times more common in *S. haemolyticus* than in other CoNS species [39]. The percentage of methicillin resistance in *S. epidermidis* (MRSE) identified in the present study (61.9%) is still lower than the 85.7% methicillin resistance identified in the mentioned study from Brazil. Although differences in occurrence and ABR between the various CoNS species are known, they are often neglected, both in studies and in clinical practice. As an example, the Dutch national report on ABR and antibiotic consumption, NethMap, combines all CoNS species into one category making it impossible to distinguish between species. Nonetheless, Dutch treatment guidelines are based on NethMap [40]. As an example, in 2019 NethMap reported for isolates found on ICUs 0% linezolid resistance in CoNS, 8% rifampicin resistance, and more than 20% resistance in all other antibiotic classes in 2019. These results could be confirmed in the present study as average values on the group level. However, these values do not reflect the true extent of resistance stratified for the single species. The lack of acknowledging ABR differences within species might cause the development of treatment guideline, and the subsequent future treatment of BSI caused by CoNS, to focus on ‘ABR-safe’ agents for treating CoNS, such as vancomycin or linezolid. Still, agents such as tetracycline, co-trimoxazole, and erythromycin could be considered viable options for some species where, according to our results, ABR never surpassed 10%. Furthermore, as age was shown to have a significant effect on ABR (Figure 4), treatment guidelines could also be improved by incorporating age-specific recommendations. We could not find the correlation between ABR in CoNS species and age in previous literature.

In the present study, some CoNS species are noteworthy to be highlighted. For instance, *S. pettenkoferi* was found only two to three times per year between 2013 and 2017 while this increased to 13 and 22 times per year in 2018 and 2019, respectively. Although recently named [41], multiple case studies showed that *S. pettenkoferi* was found to be the causative agent of septic shock, bacteraemia, and wound infections and it has also shown resistance to linezolid [41,42,43,44]. Opposingly, no linezolid resistance was found in the present study. Cases of BSI caused by *S. pettenkoferi* could incorrectly be assigned to *S. capitis* that greatly resembles *S. pettenkoferi* [41]. The emerging neonatal pathogen *S. capitis* is another noteworthy species causing sepsis and manifesting as a multidrug-resistant microorganism [45]. In this study, 7.53% of all first weighted *S. capitis* isolates was found in one-year old children. Clinically relevant ABR (e.g., to chloramphenicol or vancomycin) was not found in these children in this study. This implies that the internationally emerging *S. capitis* NRCS-A clone [45] has not been found in the Northern Netherlands between 2013 and 2019. Concerning its pathogenicity, *S. lugdunensis* has a special position among CoNS because infections due to this species resemble those caused by *S. aureus* rather than “classical” infections by CoNS [46]. Thus, it is of particular importance to know the specific AMR pattern for this species.

Our study has limitations, mostly due to its sole source of routine diagnostic data. Firstly, it was not known which isolates were causal to BSI. This hinders the assessment of contamination as well as the determination of clinical importance. Secondly, the VITEK 2 systems of the laboratories involved used different cartridges with different antibiotics, which could lead to an incorporation bias towards some laboratories or hospitals. Additionally, the MALDI-TOF MS systems of all laboratories keep their taxonomic reference data, which is proprietary, and the taxonomic recency could not be assessed. Thirdly, no molecular typing was available for any of the included isolates since genotyping was not part of routine diagnostic workflows at the time of the study. For this reason, no assessment could be made about a hospital-associated cluster of strains. Lastly, vancomycin resistance might have been underdiagnosed in this study since Vitek2 AST is not optimal for testing glycopeptide resistance [47].

For the first time, a multi-year, full-region approach to comprehensively assess both the occurrence and ABR patterns of CoNS species based on MALDI-TOF MS results was carried out. Although non-*S. lugdunensis* CoNS mainly lack aggressive virulence properties, evaluating the occurrence and ABR patterns remains highly relevant due to their impact as causative agents of foreign body-associated bloodstream infections. Stratification by region and demography unveiled a large heterogeneity in ABR between species, settings, and age groups, which could be used for (re-)evaluating treatment policies and understanding more about these important but still too often neglected pathogens.

## Figures and Tables

**Figure 1 microorganisms-10-01801-f001:**
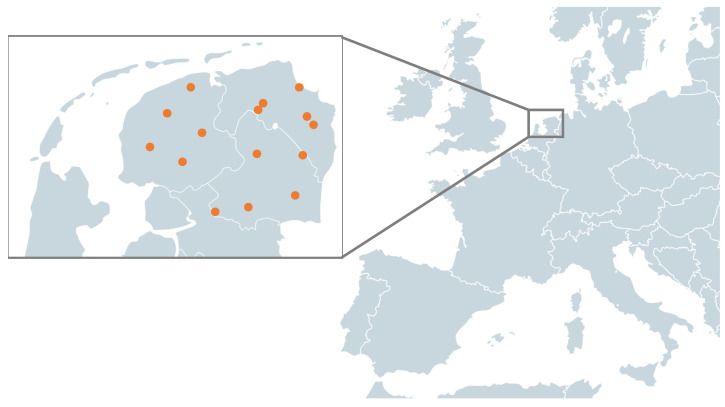
Locations of the fifteen hospitals in the three provinces in the North of the Netherlands. Between 2013 and June 2018, the region comprised fourteen hospitals; in July 2018, two hospitals merged into one new hospital, leaving a total of thirteen currently active hospitals.

**Figure 2 microorganisms-10-01801-f002:**
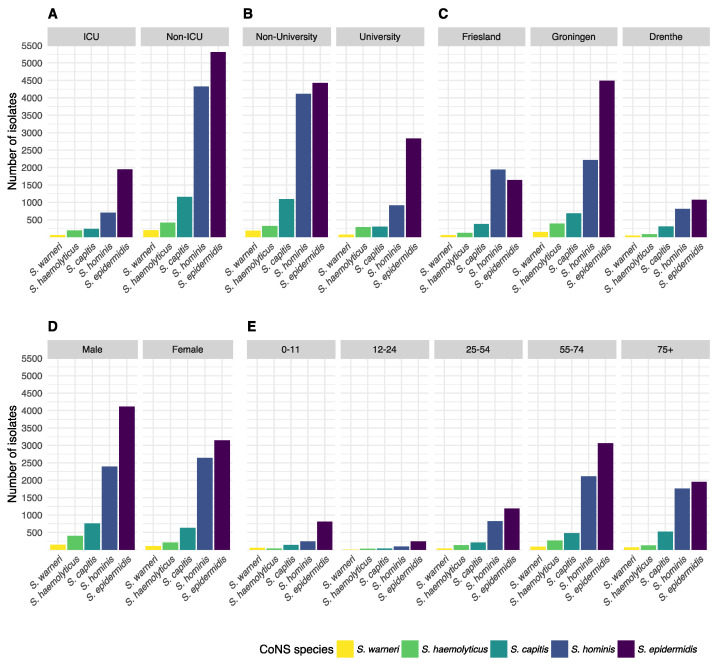
The number of first weighted isolates of the top five CoNS species found in the study stratified by (**A**) type of care, (**B**) type of hospital ward, (**C**) province of the Netherlands, (**D**), gender, and (**E**) age group.

**Figure 3 microorganisms-10-01801-f003:**
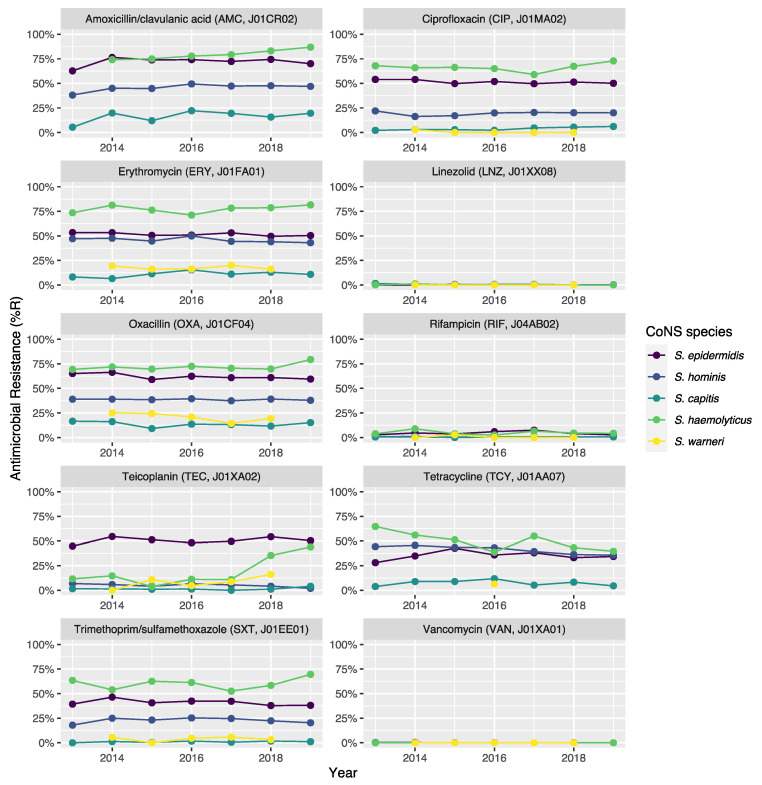
Antibiotic resistance of the five most occurrent CoNS (n = 14,560) over time between 2013 and 2019. Lines and points are missing where there were less than 30 isolates available for analysis.

**Figure 4 microorganisms-10-01801-f004:**
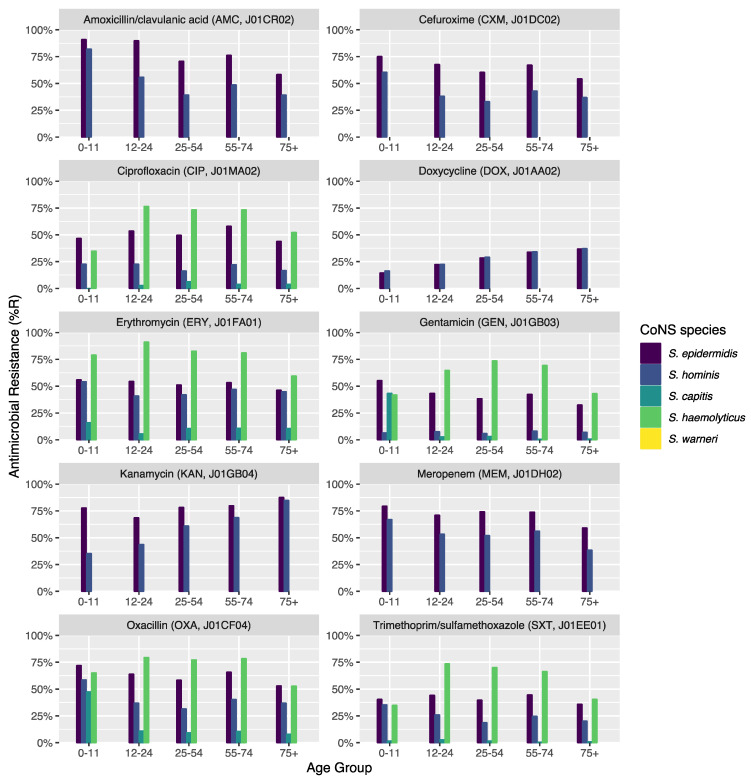
Age group comparison of ABR per antibiotic. Only microorganism-drug combinations are shown where at least 30 isolates were available for each age group and where results for all age groups were available.

**Table 1 microorganisms-10-01801-t001:** Numbers and characteristics per gender of included patients of the included CoNS isolates.

	Female	Male	Total	*p*-Value
Number of patients	5040 (46.7%)	5746 (53.3%)	10,786	<0.001
Median age	68 (IQR: 52–79)	67 (IQR: 52–77)	67 (IQR: 52–78)	<0.001
Total number of isolates	8794 (44.4%)	11,009 (55.6%)	19,803	<0.001
Number of first isolates	6026 (46.5%)	6945 (53.5%)	12,971	<0.001
Number of first weighted isolates	6887 (45.9%)	8105 (54.1%)	14,992	<0.001

**Table 2 microorganisms-10-01801-t002:** Overview of the total number of isolated CoNS species (not only first isolates) found between 2013 and 2019 in the Northern Netherlands.

Microorganism	2013	2014	2015	2016	2017	2018	2019	Total
*S. arlettae*	NF	NF	1 (0.0%)	NF	NF	NF	NF	1 (0.0%)
*S. auricularis*	3 (0.2%)	8 (0.3%)	NF	3 (0.1%)	7 (0.2%)	3 (0.1%)	6 (0.2%)	30 (0.2%)
*S. capitis*	163 (8.2%)	211 (8.0%)	259 (9.1%)	235 (7.8%)	240 (8.1%)	262 (8.2%)	276 (8.8%)	1646 (8.3%)
*S. caprae*	6 (0.3%)	5 (0.2%)	4 (0.1%)	16 (0.5%)	1 (0.0%)	11 (0.3%)	6 (0.2%)	49 (0.2%)
*S. carnosus*	1 (0.1%)	NF	2 (0.1%)	NF	NF	NF	NF	3 (0.0%)
*S. chromogenes*	NF	1 (0.0%)	NF	NF	2 (0.1%)	NF	NF	3 (0.0%)
*S. cohnii*	9 (0.5%)	6 (0.2%)	7 (0.2%)	11 (0.4%)	5 (0.2%)	5 (0.2%)	4 (0.1%)	47 (0.2%)
*S. condimenti*	NF	NF	NF	NF	NF	2 (0.1%)	1 (0.0%)	3 (0.0%)
*S. epidermidis*	1024 (51.5%)	1365 (51.7%)	1557 (54.5%)	1546 (51.2%)	1544 (51.9%)	1606 (50.3%)	1543 (49.2%)	10,185 (51.4%)
*S. equorum*	NF	NF	NF	1 (0.0%)	3 (0.1%)	NF	1 (0.0%)	5 (0.0%)
*S. gallinarum*	NF	NF	NF	1 (0.0%)	NF	NF	NF	1 (0.0%)
*S. haemolyticus*	72 (3.6%)	130 (4.9%)	141 (4.9%)	154 (5.1%)	145 (4.9%)	198 (6.2%)	141 (4.5%)	981 (5.0%)
*S. hominis*	631 (31.8%)	816 (30.9%)	789 (27.6%)	945 (31.3%)	934 (31.4%)	1009 (31.6%)	1037 (33.1%)	6161 (31.1%)
*S. kloosii*	NF	NF	NF	1 (0.0%)	NF	NF	NF	1 (0.0%)
*S. lentus ^note1^*	NF	NF	NF	1 (0.0%)	NF	1 (0.0%)	1 (0.0%)	3 (0.0%)
*S. lugdunensis*	27 (1.4%)	25 (0.9%)	11 (0.4%)	19 (0.6%)	17 (0.6%)	23 (0.7%)	31 (1.0%)	153 (0.8%)
*S. massiliensis*	NF	4 (0.2%)	NF	NF	NF	NF	NF	4 (0.0%)
*S. pasteuri*	3 (0.2%)	4 (0.2%)	6 (0.2%)	3 (0.1%)	3 (0.1%)	4 (0.1%)	9 (0.3%)	32 (0.2%)
*S. pettenkoferi*	3 (0.2%)	2 (0.1%)	3 (0.1%)	2 (0.1%)	2 (0.1%)	13 (0.4%)	22 (0.7%)	47 (0.2%)
*S. piscifermentans*	NF	NF	NF	NF	NF	1 (0.0%)	2 (0.1%)	3 (0.0%)
*S. saccharolyticus*	NF	1 (0.0%)	4 (0.1%)	1 (0.0%)	5 (0.2%)	2 (0.1%)	5 (0.2%)	18 (0.1%)
*S. saprophyticus*	2 (0.1%)	12 (0.5%)	7 (0.2%)	16 (0.5%)	10 (0.3%)	6 (0.2%)	3 (0.1%)	56 (0.3%)
*S. schleiferi ^note2^*	7 (0.4%)	6 (0.2%)	17 (0.6%)	7 (0.2%)	4 (0.1%)	3 (0.1%)	4 (0.1%)	48 (0.2%)
*S. sciuri ^note1^*	NF	1 (0.0%)	1 (0.0%)	2 (0.1%)	NF	NF	NF	4 (0.0%)
*S. simulans*	4 (0.2%)	1 (0.0%)	3 (0.1%)	4 (0.1%)	4 (0.1%)	6 (0.2%)	6 (0.2%)	28 (0.1%)
*S. warneri*	31 (1.6%)	39 (1.5%)	42 (1.5%)	49 (1.6%)	46 (1.5%)	38 (1.2%)	38 (1.2%)	283 (1.4%)
*S. xylosus*	1 (0.1%)	2 (0.1%)	1 (0.0%)	NF	3 (0.1%)	NF	1 (0.0%)	8 (0.0%)
**Total**	**1987 (100%)**	**2639 (100%)**	**2855 (100%)**	**3017 (100%)**	**2975 (100%)**	**3193 (100%)**	**3137 (100%)**	**19,803 (100%)**

Between parentheses: ratio in that year. NF = Not found. Note 1: *S. lentus* and *S. sciuri* have been reclassified to the genus *Mammaliicoccus* after the study period. Note 2: The subspecies *S. schleiferi coagulans* has been reclassified to the coagulase-positive *S. coagulans* after the study period. It was unknown which of these *S. schleiferi* isolates were of this subspecies.

**Table 3 microorganisms-10-01801-t003:** CoNS isolates found in the participating university hospital, comparing the number of first isolates found in Haematology wards vs. other wards.

Microorganism	Isolates from Haematology Ward	Isolates from Other Wards	Total Number of Isolates	Percentage from Haematology Wards	*p* Value
*S. arlettae*	0	1	1	0.0%	*p* = 1
*S. auricularis*	0	2	2	0.0%	*p* = 1
*S. capitis*	11	292	303	3.6%	***p* < 0.001**
*S. caprae*	1	13	14	7.1%	*p* = 0.33
*S. cohnii*	2	7	9	22.2%	*p* = 1
*S. condimenti*	0	2	2	0.0%	*p* = 1
*S. epidermidis*	617	2217	2834	21.8%	*p* = 0.41
*S. equorum*	0	1	1	0.0%	*p* = 1
*S. haemolyticus*	154	138	292	52.7%	***p* < 0.001**
*S. hominis*	174	745	919	18.9%	***p* = 0.047**
*S. lugdunensis*	4	21	25	16.0%	*p* = 0.63
*S. massiliensis*	0	1	1	0.0%	*p* = 1
*S. pasteuri*	2	8	10	20.0%	*p* = 1
*S. pettenkoferi*	0	15	15	0.0%	*p* = 0.052
*S. saccharolyticus*	0	3	3	0.0%	*p* = 1
*S. saprophyticus*	1	11	12	8.3%	*p* = 0.48
*S. schleiferi*	1	8	9	11.1%	*p* = 0.69
*S. sciuri*	0	2	2	0.0%	*p* = 1
*S. simulans*	0	10	10	0.0%	*p* = 0.13
*S. warneri*	3	69	72	4.2%	***p* < 0.001**
*S. xylosus*	0	2	2	0.0%	1

**Table 4 microorganisms-10-01801-t004:** The number of patients with and without CoNS persistence per species.

Microorganism	Patients with CoNS Persistence	Patients without CoNS Persistence	Total Number of Patients	Percentage of Patients with CoNS Persistence	Comparison with Other Species *
*S. capitis*	5	1351	1356	0.4%	*p* < 0.001
*S. epidermidis*	212	5466	5678	3.7%	*p* < 0.001
*S. haemolyticus*	32	519	551	5.8%	*p* < 0.001
*S. hominis*	40	4496	4536	0.9%	*p* < 0.001
*S. lugdunensis*	3	86	89	3.4%	*p* = 0.46
*S. massiliensis*	1	0	1	100%	N/A
*S. saprophyticus*	1	42	43	2.3%	*p* = 1
All other species	0	547	547	0%	N/A

N/A: not applicable (not all values > 0). Please note that this table represents number of patients, not the number of isolates. * Fisher Exact Test on 2 × 2 contingency tables comparing number of patients with CoNS persistence with current species vs. all other species.

**Table 5 microorganisms-10-01801-t005:** Antibiotic resistance in all first weighted CoNS isolates in blood between 2013 and 2019 where at least 30 isolates were available for ABR data analysis. Resistance of 100% denotes intrinsic resistance, as defined by EUCAST. Between parentheses are the number of resistant first weighted isolates and the total number of first weighted isolates for that microorganism-drug combination. The antibiotic names are followed by the official EARS-Net code (European Antimicrobial Resistance Surveillance Network) and ATC code (Anatomical Therapeutic Chemical).

Antibiotic Class	Antibiotic	*S.* *capitis*	*S.* *caprae*	*S.* *cohnii*	*S.* *epidermidis*	*S.* *haemolyticus*	*S.* *hominis*	*S.* *lugdunensis*	*S.* *pettenkoferi*	*S.* *saprophyticus*	*S.* *schleiferi*	*S.* *warneri*
Aminoglycosides	Gentamicin (GEN, J01GB03)	5.8% (71/1232)	2.6% (1/39)	0.0% (0/38)	40.7% (2744/6739)	62.7% (352/561)	7.3% (333/4553)	3.7% (3/81)	0.0% (0/37)	0.0% (0/39)	0.0% (0/34)	13.2% (30/228)
	Kanamycin (KAN, J01GB04)	34.3% (79/230)			80.3% (3174/3954)	89.7% (358/399)	67.0% (819/1222)					41.9% (31/74)
	Tobramycin (TOB, J01GB01)	6.6% (79/1198)	5.1% (2/39)	2.9% (1/35)	47.4% (3167/6686)	63.1% (355/563)	18.1% (816/4518)	3.7% (3/82)	0.0% (0/35)	0.0% (0/38)	0.0% (0/34)	13.0% (30/230)
Amphenicols	Chloramphenicol (CHL, J01BA01)	0.4% (3/817)			1.6% (75/4662)	4.4% (17/386)	2.7% (88/3291)	0.0% (0/53)	0.0% (0/30)	0.0% (0/33)		0.6% (1/170)
Antimycobacterials	Rifampicin (RIF, J04AB02)	0.6% (7/1218)	0.0% (0/39)	0.0% (0/37)	4.5% (300/6709)	5.0% (28/562)	0.5% (25/4549)	0.0% (0/82)	2.8% (1/36)	0.0% (0/39)	0.0% (0/34)	0.4% (1/229)
Beta-lactams	Amoxicillin/ clavulanic acid (AMC, J01CR02)	16.6% (107/643)			72.9% (3026/4153)	77.8% (280/360)	46.1% (1064/2310)	11.9% (5/42)				30.5% (36/118)
	Benzylpenicillin (PEN, J01CE01)	78.9% (870/1102)	83.3% (30/36)	97.1% (34/35)	92.5% (5927/6411)	89.2% (486/545)	88.2% (3714/4209)	57.5% (42/73)	85.3% (29/34)		25.0% (8/32)	77.8% (161/207)
	Oxacillin (OXA, J01CF04)	13.4% (160/1197)	5.1% (2/39)	64.7% (22/34)	61.9% (4135/6679)	72.1% (403/559)	38.6% (1692/4386)	7.3% (6/82)	40.0% (14/35)	16.2% (6/37)	0.0% (0/34)	20.5% (47/229)
Carbapenems	Cefoxitin (FOX, J01DC01)	21.8% (67/307)			73.4% (1823/2484)	79.8% (197/247)	54.3% (506/932)					28.6% (16/56)
Glycopeptides	Teicoplanin (TEC, J01XA02)	1.4% (17/1190)	0.0% (0/39)	0.0% (0/33)	50.5% (2752/5448)	20.2% (105/520)	5.1% (202/4000)	3.7% (3/82)	0.0% (0/31)	0.0% (0/38)		9.6% (22/230)
	Vancomycin (VAN, J01XA01)	0.0% (0/1195)	0.0% (0/39)	0.0% (0/35)	0.1% (6/6674)	0.0% (0/560)	0.0% (1/4524)	0.0% (0/82)	0.0% (0/35)	0.0% (0/38)	0.0% (0/34)	0.0% (0/230)
Macrolides/ lincosamides	Azithromycin (AZM, J01FA10)	11.0% (136/1232)	12.8% (5/39)	44.7% (17/38)	51.5% (3471/6741)	77.6% (437/563)	45.7% (2087/4564)	12.2% (10/82)	18.9% (7/37)	15.4% (6/39)	0.0% (0/34)	17.5% (40/228)
	Clarithromycin (CLR, J01FA09)	11.0% (136/1231)	12.8% (5/39)	44.7% (17/38)	51.5% (3471/6741)	77.6% (437/563)	45.7% (2086/4563)	12.2% (10/82)	18.9% (7/37)	15.4% (6/39)	0.0% (0/34)	17.5% (40/228)
	Clindamycin (CLI, J01FF01)	10.8% (132/1224)	5.1% (2/39)	26.3% (10/38)	43.4% (2910/6706)	45.6% (253/555)	29.6% (1347/4547)	12.0% (10/83)	38.9% (14/36)	5.1% (2/39)	0.0% (0/34)	4.4% (10/229)
	Erythromycin (ERY, J01FA01)	11.0% (136/1231)	12.8% (5/39)	44.7% (17/38)	51.5% (3471/6741)	77.6% (437/563)	45.7% (2086/4563)	12.2% (10/82)	18.9% (7/37)	15.4% (6/39)	0.0% (0/34)	17.5% (40/228)
Other antibacterials	Fosfomycin (FOS, J01XX01)			48.4% (15/31)	9.7% (542/5616)	96.6% (458/474)	97.6% (3426/3509)	16.9% (11/65)				99.5% (191/192)
	Mupirocin (MUP, R01AX06)	2.6% (32/1215)	2.6% (1/39)	0.0% (0/36)	14.8% (987/6687)	3.7% (21/562)	6.5% (297/4536)	0.0% (0/82)	0.0% (0/37)	2.6% (1/39)	0.0% (0/34)	1.7% (4/230)
	Nitrofurantoin (NIT, J01XE01)	0.7% (2/280)			0.6% (10/1732)	0.0% (0/160)	0.9% (8/926)					0.0% (0/55)
Oxazolidinones	Linezolid (LNZ, J01XX08)	0.4% (5/1192)	0.0% (0/39)	0.0% (0/33)	0.1% (5/6641)	0.2% (1/559)	0.4% (17/4508)	0.0% (0/82)	0.0% (0/34)	7.9% (3/38)	0.0% (0/34)	0.0% (0/230)
Quinolones	Ciprofloxacin (CIP, J01MA02)	3.8% (47/1229)	2.6% (1/38)	0.0% (0/38)	51.5% (3468/6740)	66.4% (374/563)	19.4% (885/4561)	4.8% (4/83)	8.1% (3/37)	0.0% (0/39)	0.0% (0/34)	0.4% (1/229)
	Moxifloxacin (MFX, J01MA14)	0.0% (0/396)			6.8% (88/1292)	26.4% (24/91)	3.6% (45/1239)	0.0% (0/30)				0.0% (0/88)
Tetracyclines	Doxycycline (DOX, J01AA02)	5.9% (54/914)			32.9% (1182/3597)	46.9% (122/260)	34.0% (967/2848)	14.5% (9/62)			0.0% (0/30)	3.5% (6/172)
	Minocycline (MNO, J01AA08)	0.0% (0/826)			0.0% (1/2496)	0.6% (1/158)	0.3% (6/2064)	0.0% (0/53)			0.0% (0/30)	0.0% (0/168)
	Tetracycline (TCY, J01AA07)	7.3% (71/971)		6.7% (2/30)	35.0% (1521/4347)	50.2% (163/325)	41.0% (1544/3765)	15.4% (10/65)			0.0% (0/30)	4.5% (8/176)
Trimethoprim and combinations	Trimethoprim (TMP, J01EA01)	10.6% (36/339)			77.4% (2923/3775)	90.2% (350/388)	55.2% (1364/2470)					17.5% (10/57)
	Trimethoprim/ sulfamethoxazole (SXT, J01EE01)	1.0% (12/1233)	0.0% (0/39)	2.6% (1/38)	41.0% (2764/6735)	60.1% (338/562)	22.8% (1037/4556)	2.4% (2/82)	2.7% (1/37)	0.0% (0/39)	0.0% (0/34)	3.5% (8/229)

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
