# Peer review of "Trends in Occurrence and Phenotypic Resistance of Coagulase-Negative Staphylococci (CoNS) Found in Human Blood in the Northern Netherlands between 2013 and 2019"

_microorganisms, 2022, doi:10.3390/microorganisms10091801_

Round 1

Reviewer 1 Report

This study entitled "" was carried out to determine a multi-year, full-region approach to comprehensively assess both occurrence and ABR patterns of CoNS species based on MALDI-TOF, however the following points should be addressed:

-Did authors calculate sample size prior sampling from patients.

-One limitation for this study that different Vitek cartridge were used for testing antimicrobial susceptibility. It is recommended to use the same cartridge for for testing isolates from different districts, labs to the same panel of antimicrobials.

-More details about how AMR package, R software, used to exclude and select isolates for their antimicrobial resistance patterns, should be given.

Reviewer 2 Report

Authors provide extensive retrospective study on antibiotic resistance of  CoNS isolated from blood cultures in 15 hospitals within the Northern Netherlands. It highlights the differentiation of this particular group of staphylococci and the resistance to species level. Although the study has some limitations as part of the routine work in the hospitals, it has laboratory and clinical  significance.

Suggestions:

- In the current title the origin of blood is not specified; for readers is important to understand at glans if the study would refer to human or animal blood cultures; please specify

Keywords: coagulase-negative staphylococci (CoNS); antimicrobial resistance; Staphylococcus haemolyticus; Staphylococcus epidermidis; blood, humans, epidemiology; trends
